# The role of genetic essentialism and genetics knowledge in support for eugenics and genetically modified foods

Benjamin Y. Cheung, Anita Schmalor, Steven J. Heine *

University of British Columbia, Vancouver, Canada

* heine@psych.ubc.ca

## Abstract

People are regularly exposed to discussions about the role of genes in their lives, despite often having limited understanding about how they operate. The tendency to oversimplify genetic causes, and ascribe them with undue influence is termed genetic essentialism. Two studies revealed that genetic essentialism is associated with support for eugenic policies and social attitudes based in social inequality, and less acceptance of genetically modified foods. These views about eugenics and genetically-modified foods were especially evident among people who had less knowledge about genes, potentially highlighting the value of education in genetics.

## Introduction

There is much consensus that the eugenics movement of the early 20th century was one of the most problematic applications of scientific theorizing towards political ends [e.g., 1–3]. The key goal of the eugenics movement was to improve the species-wide human genome for future generations. At first eugenic policies were restricted to so-called "positive eugenics" efforts in which people with desired qualities were encouraged to reproduce [2–4]. Soon, however, this changed into extensive programs of negative eugenics in which people with undesired qualities were actively discouraged and prevented from reproducing. As a result of these efforts, numerous laws were passed in such diverse places as the US, Canada, Japan, Scandinavia, and much of Latin America (some of which remained in place until the late 1970's), that resulted in the forceful sterilization of hundreds of thousands of individuals across the world [5–8]. Most disturbingly, millions of people were murdered in Nazi Germany for concerns that they would pass the wrong "essence" on to future generations.

Given the deeply disturbing past of eugenics it is not surprising that the idea is so widely despised today. However, to get a better understanding of how eugenic ideas became so widely implemented, it is worth revisiting just how popular the movement was in the early part of the 20th century. At this time, eugenic policies had extremely broad support, and were championed by leading thinkers on both sides of the political continuum [9–11]. It is perhaps surprising today to learn that some of the most ardent eugenic supporters included famous intellectuals such as Alexander Graham Bell, H. G. Wells, George Bernard Shaw, that

www.sshrc-crsh.gc.ca/home-accueil-eng.aspx No
- The funders had no role in study design, data
collection and analysis, decision to publish, or
preparation of the manuscript.

**Competing interests:** The authors have declared
that no competing interests exist.

it received generous funding from families of Carnegie, Eastman, Guggenheim, Kellogg, Rockefeller, and Vanderbilt [8, 12–14], and that by 1928 a total of 376 different universities offered courses on it [15]. While many of the founding fathers of psychology were active members in the eugenics movement, including Carl Brigham, James McKeen Cattell, Robert Fisher, G. Stanley Hall, Karl Pearson, Charles Spearman, Lewis Terman, Edward Thorndike, and Robert Yerkes [2, 3, 8], genetics was the field that was most closely associated with eugenics. In the early 20th century, eugenics was largely thought of as applied human genetics [3]. For example, in 1916 every member of the founding editorial board of the journal *Genetics* endorsed the eugenics movement [16]. Likewise, the first professional genetics society was the American Breeders Associated, established in 1903, which was largely concerned with improving the genetic stock of both livestock and humans [9]. It is also notable that more than half of all academic biologists were members of the Nazi party, the largest representation of any professional group [3]. These parallels suggest that while many have critiqued the arguments for genetic causes put forth by eugenicists as overly simplistic and deterministic [e.g., 2, 9], in the early days of genetics research such views were not particularly uncommon [for a review, see 17].

Attitudes towards eugenic ideas became sharply more negative after World War 2, as it became evident just how horrific were the methods that the Nazis had pursued towards eugenic goals. After this period, discussions of eugenics quickly became verboten, and several eugenic societies and journals adopted name changes to conceal their association with this disturbing movement [18]. One might have hoped that the history of eugenics would have permanently ended there. But, as some have argued, the eugenic goals of improving the human genome have not really disappeared. With the advent of new genetic engineering technologies, some have suggested that we're returning to eugenics through the back door [19, 20].

An idea with such dangerous implications as eugenics deserves to be better understood. Advances in new genetic engineering technologies, such as preimplantation genetic diagnosis and CRISPR-Cas9-based genome-editing, have arguably advanced faster than our abilities to make sense of their associated ethical implications [21, 22]. In this paper we strive to explore the kinds of attitudes towards genetics that predict support for eugenic policies.

Relatedly, in this paper we also consider people's attitudes towards another effort to modify genomes–that of genetically-modified organisms (GMO). While humans have been indirectly modifying the genomes of their crops since the first agricultural plants were domesticated, in recent years this modification has become more direct, as new GMO technologies allow novel genes to be directly injected into the cells of other organisms. GMO food products have rapidly entered our diets–one estimate is that as much of 80% of processed food in the US contains some GMO content [23]. Despite their prevalence, GMO technologies remain quite controversial: a New York Times poll from 2013 found that 3/4 of Americans expressed concerns about consuming GMOs, with a fear of health consequences being the most commonly cited worry [24]. Research finds that opposition to GMO products is not only common, but a majority of opponents of GMO oppose it in absolute terms–viewing it as morally wrong, regardless of the particular risks or benefits involved [25]. Moreover, research finds that those with the most extreme opposition to GMO foods tend to have lower knowledge about science and genetics, despite being confident in their perceived understanding of GMO foods [26]. However, confidence in opposition to GMO products diminishes when people are pressed for their actual knowledge of what is involved in genetic modification [27]. We also seek to better understand people's attitudes towards GMO products in the context of their beliefs about genetic causes more generally.

## Genetic essentialism

People often struggle with learning complex scientific concepts. In the case of genetics, this problem is exacerbated because people have intuitions that may mislead them in their understanding. These intuitions relate to psychological essentialism–a tendency for people to understand the natural world as emerging from some deep, internal, and hidden forces. Such forces, or *essences*, serve as the basis of natural categories [28–30]. Because people have a difficult time forming a concrete mental representation of essences it has been argued that they rely on essence placeholders which serve as a scaffolding for explaining how the natural world came to be [31]. A particularly compelling placeholder is the lay person's understanding of genes. As is the case with essences, people view genes as being the ultimate explanation for the origin of an organism, and, like essences, one's genes are present at the moment of conception, and are largely unchanged across one's life. This tendency to understand genes by way of our intuitions regarding essences, has been termed genetic essentialism [32, 33].

Dar-Nimrod and Heine [32] argue that there are four key aspects of genetic essentialist thinking. First, genetic causes tend to be perceived as immutable; people may view that someone who possesses a gene linked to a particular condition will inevitably develop that condition. Second, when people make genetic attributions for a phenomenon, they often discount other kinds of potential causes so that genetic factors may be seen as the ultimate cause. A third feature of genetic essentialist thinking is that people tend to view groups that share genes as more homogenous, and more discrete from groups that possess alternative alleles; that is, genes are perceived to serve as the basis of natural categories. Last, genes tend to be perceived as natural causes, and, as such, any interventions to modify them may strike people as deeply unnatural, and bothersome.

Much research on genetic essentialism has revealed that when people tend to view genes in simplistic and deterministic ways, either by manipulations to make those thoughts more accessible or by comparing people who chronically tend to view genes in these ways, they have a number of predictable responses [for a review, see [34]]. For example, when people hear of a genetic cause of criminal behavior they tend to view perpetrators as less responsible for their crimes and as more dangerous [35]; and when people reflect on the role that genes have in obesity they come to think of one's weight as beyond one's control [36]. Likewise, encounters with genetic arguments for sexual orientation lead people to be more accepting of same-sex marriage [37], whereas genetic arguments for sex differences in math performance lead women to perform worse on math tests [38]. Genetic attributions for mental illness lead people to be less likely to blame individuals for their condition [39, 40], however, they also are associated with more pessimistic prognoses [41, 42]. Moreover, people who have more genetic essentialist perspectives in general tend to score higher on measures of racism and sexism [43, 44].

Given that simplistic views of genetic causes influence the ways that people think about so many phenomena, how might genetic essentialism relate to people's support for eugenics? Some research has found that people do express concern about who reproduces when genetic causes are discussed. For instance, when mental health conditions are portrayed as having genetic causes, concerns about mental health status extend to the offspring of someone with the condition [40]. Similarly, perceiving intelligence as primarily genetically based is associated with a tendency to view existing racial differences in intelligence as immutable, and to regard it as a trait that passes down through generations [45]. Moreover, a look back at some attitudes of leading figures in the eugenics movement of the 20th century may be informative. In the early years of the eugenics movement, human conditions were often described as though they emerged from a single gene. For example, one of the leading American eugenicists, Charles B. Davenport, proposed that a long list of curious human conditions, such as "nomadism,"

"shiftlessness," "innate eroticism," and thalassophilia ("love of the sea") were the product of monogenic causation [2, 9]. In the US, and much of the world, eugenic policies were adopted to combat "feeblemindedness," a trait that described one's intellectual and moral character [18]. Many viewed this trait to be the product of a single gene, and the logic seemed to follow that if one could decrease the frequency of this gene, then one could improve the nation's intellectual and moral character [46, 47]. That simple deterministic genetic accounts for human conditions were common when eugenics beliefs were widespread raises the possibility that people who conceive of genetics in essentialist ways may be more convinced by arguments for eugenics policies.

Similarly, we can also consider how essentialist views about genetics might relate to attitudes towards GMOs. While opposition to GMOs is widespread, it is concentrated within particular populations. One survey found that only 37% of American adults were open to GMOs (in contrast to 88% of scientists; [48]; also see [49, 50]). While concerns with GMOs generalize beyond fears of the genes themselves (e.g., people may be bothered by the resulting homogenization of the food supply, the increased use of herbicides, the centralized role of large agribusinesses, the effects on the ecosystem, etc.), it is instructive to consider this widespread opposition to GMOs alongside common misunderstandings of what GMOs entail. For example, one study found that only 57% of Americans were aware that "ordinary tomatoes contain genes" [51]. Similarly, 20% of Europeans endorsed a common misconception about GMOs that a person's own genes may change if they consume GMO food [52]. Indeed, more than half of American adults acknowledged that they knew "very little or nothing at all" about GMOs [53]. The ways that our food is genetically modified is not readily apparent to many lay people and the inherent complexity of this topic may relate to people's fears about GMOs. We wished to also explore whether rather simplistic understandings about genes related to attitudes towards GMOs.

## The present research

We conducted two correlational studies to explore how attitudes towards eugenics and GMOs related to essentialist understandings about genes. In Study 1 we explored the relations between genetic essentialism and support for eugenics. We also included a measure of social dominance orientation as past work has found that this relates to genetic essentialism [43], and it seems relevant to attitudes towards eugenic policies. In Study 2 we preregistered an effort to replicate the pattern from Study 1, and further examined people's support for GMOs. In addition, in Study 2 we investigated how knowledge about genetics related to people's support for eugenics and GMOs.

## Study 1

### Methods

**Participants.**   This first exploratory study had 1350 participants, involving 1084 American workers from MTurk, and 266 undergraduate students from the University of British Columbia. Of these participants, 586 identified as men, 709 identified as women, 7 chose "other", while the remaining did not disclose their gender. The participants have a mean age of 31.94, *SD* = 11.98.

**Materials.**   Participants completed the following self-report measures, which were analyzed to discern the correlations among them:

*Belief in Genetic Determinism*. Participants completed the 18-item Belief in Genetic Determinism Scale [43] on a 7-point scale ranging from 1 (Strongly Disagree) to 7 (Strongly Agree). Sample items include "In my opinion, alcoholism is caused primarily by genetic factors" and

"The fate of each person lies in his or her genes." This scale is a widely used measure that assesses the degree to which people believe various human characteristics are the direct product of inferred underlying genetic differences [e.g., 54]. Cronbach's α = 0.96.

*Genetic Essentialist Tendencies.* Participants completed the 24-item Genetic Essentialist Tendencies Scale [55] on a 5-point scale ranging from 1 (Strongly Disagree) to 5 (Strongly Agree). Sample items include "People with a genetic predisposition to be intelligent eventually will show intellectual achievements" and "The environment does not affect the changes of getting cancer for someone with a genetic susceptibility to cancer." (see S1 Appendix for the complete list of items and means). This scale builds from the genetic essentialist framework from [32] and assesses the degree to which people view genetic causes in essentialist terms. Previous research suggests it correlates strongly with the Belief in Genetic Determinism measure [e.g., 56]. Cronbach's α = 0.93. We included two distinct measures of genetic essentialist biases (the Belief in Genetic Determinism Scale and the Genetic Essentialist Tendencies Scale) as a test of robustness.

*Social dominance orientation.* Participants completed the 16-item Social Dominance Orientation Scale [57] on a 7-point scale ranging from 1 (Strongly Oppose) to 7 (Strongly Favor). Sample items include "Some groups of people must be kept in their place" and "We should not push for group equality." This widely used scale assesses the degree to which people view some social groups to be superior to others. Cronbach's α = 0.95.

*Eugenics acceptance.* Participants completed the 15-item measure of eugenics acceptance that was created for this study on a 5-point scale ranging from 1 (Strongly Disagree) to 5 (Strongly Agree). Sample items include "Sterilization of those possessing undesirable traits (e.g., a disorder) is a way to improve future generations" and "People with a criminal record should be prevented from having biological children" (see S2 Appendix for the complete list of items and means). Cronbach's α = 0.93.

This study, as well as Study 2, were approved by the University of British Columbia Behavioral Research Ethics Board (H14-00250); written consent was obtained.

## Results and discussion

We correlated people's attitudes towards eugenic acceptance and social dominance orientation with two measures of genetic essentialist biases: Beliefs in Genetic Determinism, and the Genetic Essentialism Tendencies scale.

On average, people scored below the midpoint of the scale on eugenic acceptance, indicating that there was not a great deal of endorsement of these items (see Table 1). Nonetheless, all of the measures correlated with eugenic acceptance. Younger people and men tended to endorse more support for eugenic policies than older people and women. Likewise, younger people and men scored higher on a social dominance orientation. The two measures of genetic essentialist biases were highly correlated with each other, $r = .60$, $p < .001$. Replicating [43] we found that people's social dominance orientation was significantly predicted by Beliefs in Genetic Determinism, $r = .17$, as well as by Genetic Essentialist Tendencies, $r = .33$, both $ps < .001$. Likewise, we also found that people's attitudes towards eugenics were significantly predicted by both measures of genetic essentialist biases, $rs = .27$ and .39, $p < .001$, for Beliefs in Genetic Determinism and Genetic Essentialist Tendencies, respectively. That is, people who had more essentialist views of genetics tended to view some social groups as superior to others and they tended to have more support for state-sanctioned eugenic policies.

We wished to replicate and extend these findings. First, we assessed whether the same correlations between eugenic acceptance, social dominance orientation, and genetic essentialist beliefs could be replicated in a preregistered analysis. Second, we explored whether people's

**Table 1. Means, alphas, and bivariate correlations for all variables in Study 1.**

| Variables | Mean (SD) | α | 1 | 2 | 3 | 4 | 5 | 6 |
|---|---|---|---|---|---|---|---|---|
| 1. Beliefs in Genetic Determinism<br>1–7 scale from "Not at all true" to "Completely True" | 4.19 (0.78) | 0.86 | 1.00 | | | | | |
| 2. Genetic Essentialist Tendencies<br>1–5 scale from "Strongly Disagree" to "Strongly Agree" | 2.64 (0.61) | 0.93 | .60*** | 1.00 | | | | |
| 3. Social Dominance Orientation<br>1–5 scale from "Strongly Oppose" to "Strongly Favor" | 2.45 (1.16) | 0.95 | .17*** | .33*** | 1.00 | | | |
| 4. Eugenic Acceptance<br>1–5 scale from "Strongly Disagree" to "Strongly Agree" | 2.14 (0.77) | 0.93 | .27*** | .39*** | .41*** | 1.00 | | |
| 5. Age | 31.94 (11.98) | | .10** | .05 | -.11*** | -.15*** | 1.00 | |
| 6. Gender (Women = 0, Men = 1) | | | .04 | 0.00 | .18*** | .16*** | .05 | 1.00 |

Notes
*$p < .05$
**$p < .01$
***$p < .001$.

attitudes towards GMOs were predicted by their attitudes towards eugenics, their genetic essentialist beliefs, and their knowledge about genetics. Last, we preregistered hypotheses that people's genetics knowledge would negatively predict both measures of their essentialist tendencies, as well as their attitudes towards eugenic policies. We also asked participants how many genetics courses they had taken which we had preregistered. However, surprisingly, this measure correlated *negatively* with genetics knowledge ($r = -.22, p < .001$). For example, the more genetics courses people took, the more likely they were to say that a gene was a cell ($r = .28, p < .001$). People's answers to this item do not seem to indicate any genetics knowledge, indicating to us that people hadn't understood the question about genetic courses correctly. We also explored the role of people's political attitudes.

## Study 2

### Methods

**Participants.** As specified in the pre-registration (https://aspredicted.org/blind.php?x=gg52xv), we planned to collect data from 450 American TurkPrime workers. We aspired to have a sample size that is larger than N = 365, as according to [58], a true correlation of .10 will stabilize (ie., vary only within the corridor of stability) at a sample size of 362 when the corridor of stability is set to a half-width of .10 at a 90% confidence interval. After excluding participants according to our pre-registered criteria, we had a final sample of 398 (59% female; $M_{age}$ = 35.66, SD = 11.56).

**Materials.** Study 2 included the identical measures from Study 1 (viz., Beliefs in Genetic Determinism, Genetic Essentialist Tendencies, Social Dominance Orientation, and our measure of eugenics acceptance), and they were completed on 7-point Likert scales ranging from 1 (Strongly Disagree) to 7 (Strongly Agree). In addition, we included the following measures:

*Attitudes towards genetically-modified food.* Participants completed an 8-item measure from [25]. Five of these items asked about support of government restrictions of GMO products (e.g., "Your government forbidding imports of genetically modified (GM) foods from other countries" and participants indicated their support on a 9-point scale from 1 (Certainly oppose) to 9 (Certainly support). Three of these items asked about perceived risk of GMO products (e.g., Genetically modified foods having unknown side-effects, increasing risks of

cancer or other diseases for people who consume them" which participants answered on a 9-point scale from 1 (No risk at all) to 9 (Extreme risk) [25]. Included an additional item about GMO risks that read "Genetically modified crops giving big corporations too much power over small farmers." We omitted this item from our study as it did not seem to tap into fears of genetic modifications per se, but concerns about implications of business practices regarding GMO products. Both subscales were reverse coded so that higher scores meant more favorable attitudes towards GMO foods. This scale has been used in previous research investigating the bases of people's support and opposition to GMO foods [e.g., 59]. Cronbach's α = 0.93.

*Knowledge about genetics*. Participants completed a 21-item set of questions assessing genetics knowledge which were completed on a binary scale (see S3 Appendix for complete list of items). Eight of these items came from [60] Christensen et al., (2010) and 13 of the items came from [61] Jallinoja and Aro (1999). We counted the number of correct responses across all 21 items. These measures of genetic knowledge have been used in previous research [e.g., 56, 62].

*Political conservatism*. Participants completed two items asking their political orientation on both social issues and economic issues, which they completed on 7-point scales from 1 (Very liberal) to 7 (Very conservative). The two items were averaged.

## Results and discussion

We first calculated people's performance on the Knowledge about Genetics assessment. On average, people scored 14.12 correct out of 21 (range from 7 to 21), which is only somewhat above chance performance (10.5), indicating not a great deal of genetics knowledge among the participants.

We then calculated correlations among the different measures. As preregistered, and replicating Study 1, both measures of beliefs in genetic essentialism positively predicted social dominance, $rs$ = .41 and .55, $ps < .001$, for Beliefs in Genetic Determinism and Genetic Essentialist Tendencies, respectively (see Table 2). Likewise, as preregistered, both measures of genetic

**Table 2. Mean, alphas, and bivariate correlations for all variables in Study 2.**

| Variables | Mean (SD) | α | 1 | 2 | 3 | 4 | 5 | 6 | 7 | 8 | 9 |
|---|---|---|---|---|---|---|---|---|---|---|---|
| 1. Beliefs in Genetic Essentialism<br>1–7 scale from "Not at all true" to "Completely True" | 4.31 (0.89) | .87 | 1.00 | | | | | | | | |
| 2. Genetic Essentialist Tendencies<br>1–7 scale from "Strongly Disagree" to "Strongly Agree" | 3.20 (0.85) | .96 | .76*** | 1.00 | | | | | | | |
| 3. Social Dominance Orientation<br>1–7 scale from "Strongly Oppose" to "Strongly Favor" | 3.05 (1.21) | .91 | .41*** | .55*** | 1.00 | | | | | | |
| 4. Eugenic Acceptance<br>1–7 scale from "Strongly Disagree" to "Strongly Agree" | 2.81 (1.16) | .97 | .59*** | .79*** | .58*** | 1.00 | | | | | |
| 5. Age | 35.66 (11.56) | | -.05 | -.17** | -.15** | -.22*** | 1.00 | | | | |
| 6. Gender (Women = 0, Men = 1) | | | .17** | .25*** | .25*** | .23*** | -.05 | 1.00 | | | |
| 7. GMO Acceptance<br>1–9 scale | 3.42 (1.65) | .90 | -.26*** | -.33*** | -.17*** | -.28*** | .04 | -.01 | 1.00 | | |
| 8. Genetics Knowledge (number correct out of 21 questions) | 14.12 (2.92) | | -.35*** | -.54*** | -.40*** | -.48*** | .07 | -.20*** | .24*** | 1.00 | |
| 9. Political Conservatism<br>1–7 scale from "Very Liberal" to "Very Conservative" | 4.23 (1.91) | | .37*** | .50*** | .61*** | .51*** | -.06 | -.23*** | -.25*** | -.35*** | 1.00 |

Notes:

*$p < .05$

**$p < .01$

***$p < .001$.

essentialism positively predicted eugenics acceptance, $rs$ = .59 and .79, $ps < .001$, for Beliefs in Genetic Determinism and Genetic Essentialist Tendencies, respectively. Again, eugenics acceptance and social dominance were more favored by younger people and by men. Unexpectedly, these correlations were considerably larger than they were in Study 1, and we are not sure what aspect of the sample or survey design might be responsible for these differences in magnitude.

We also found that GMO acceptance was negatively correlated with people's support for eugenics, $r$ = -.28, $p < .001$; that is, the more people supported eugenics the more negative were their attitudes towards GMOs. Beliefs in genetic essentialism were negatively associated with GMO acceptance, $rs$ = -.26 and -.33 for Beliefs in Genetic Determinism and Genetic Essentialist Tendencies, respectively ($ps > .001$). That is the more simplistic and deterministic that people viewed genetic causes, the more they opposed GMOs.

In line with our preregistered hypotheses, the more people knew about genetics as assessed by our measure of Knowledge about Genetics, the lower they scored on measures of genetic essentialism, $rs$ = -.35 and -.54 for Beliefs in Genetic Determinism and Genetic Essentialist Tendencies, respectively ($ps > .001$). Likewise, as preregistered, those people who knew more about genetics scored lower on eugenics acceptance, $r$ = -.48. We also found that knowledge about genetics was negatively correlated with social dominance, $r$ = -.40 ($ps > .001$), indicating that the more participants knew about genetics the less likely they viewed it as appropriate for some groups to dominate others. In addition, although not preregistered, we found that knowledge about genetics was positively correlated with GMO acceptance, $r$ = .24, $p < .001$. That is, those participants who knew more about genetics tended to have more positive attitudes towards GMOs, a finding which converges with some other research [e.g., 26]. We also found that political conservatism was positively associated with genetic essentialist beliefs, social dominance, and eugenic support, but was negatively associated with attitudes towards genetically modified foods and with genetics knowledge.

## General discussion

The results of these studies show that views on controversial topics related to genetics are predicted by the ways that people think about genes. Those who essentialize genetic concepts are more likely to view some groups as being inferior to others, support government eugenic policies to control who reproduces, and have fears about GMO foods. Given that GMO foods are widely viewed as safe by scientists, and social dominance orientations and eugenic beliefs are clearly harmful, genetic essentialist views can have pronounced costs.

Genetic essentialism involves an overly simplistic determinism between genotypes and their associated phenotypes [see 34] for a review]; but genetic effects are far more nuanced and complex (for thoughtful reviews on this see [63, 64]). An underappreciation of this complexity may make eugenic ideas more appealing, leading people to condemn those who are assumed to have problematic genomes. Likewise, misunderstandings about genetics may render GMO foods as more threatening. While there are many reasons why people are cautious towards GMOs, a part of people's concerns appears to hinge on GMOs seeming to violate the natural order of things and of crossing the perceived boundaries of essences [17, 25]. People with less essentialized views about genetics appear to share fewer of these concerns.

In addition to genetic essentialism, we found that knowledge about genetics also significantly predicted people's views on these topics. It is encouraging that genetics knowledge was associated with less essentialist views, weaker support for social dominance and eugenics, and less negative attitudes towards GMO foods. This raises the possibility that some of people's more harmful attitudes may be reduced by appropriate genetics education. Indeed, several

efforts have revealed that many of the simplistic and problematic views about genetics can be reduced by curricula that consciously strive to counter them [e.g., 65–68]. The present research suggests that such efforts might help people to become more critical of eugenic and social dominance perspective [also see 69, 70], and may lead to become more open to some of the potential benefits offered by GMO technologies.

## Limitations

These conclusions are limited by the correlational nature of the data, which renders it difficult to infer causality. In particular, it is possible that a third variable, such as general trust in science, cognitive style, or broad science education, may underlie the correlations of some of these variables, including attitudes towards GMOs, eugenics acceptance, and genetics knowledge. Future research efforts would benefit by exploring experimental means to manipulate the ways that people think about genetics in order to investigate whether this leads to subsequent changes in people's attitudes towards eugenics and GMOs. Moreover, given that all of the present data were collected from North Americans it remains an open question as to whether they would generalize elsewhere. It would be informative to know whether similar attitudes towards genetics are found in other contexts.

## Supporting information

**S1 Appendix. Genetic essentialistic tendencies measure.**
(DOCX)

**S2 Appendix. Eugenics acceptance measure.**
(DOCX)

**S3 Appendix. Measure of genetics knowledge.**
(DOCX)

## Author Contributions

**Conceptualization:** Steven J. Heine.

**Formal analysis:** Benjamin Y. Cheung, Anita Schmalor.

**Funding acquisition:** Steven J. Heine.

**Methodology:** Benjamin Y. Cheung, Anita Schmalor, Steven J. Heine.

**Supervision:** Steven J. Heine.

**Writing – original draft:** Benjamin Y. Cheung.

**Writing – review & editing:** Benjamin Y. Cheung, Anita Schmalor, Steven J. Heine.

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
