## [Decision Letter · Decision Letter 0]

27 Apr 2021

PONE-D-21-03521

The Role of Genetic Essentialism and Genetics Knowledge in Support for Eugenics and Genetically Modified Foods

PLOS ONE

Dear Dr. Heine,

Thank you for submitting your manuscript to PLOS ONE. After careful consideration, we feel that it has merit but does not fully meet PLOS ONE’s publication criteria as it currently stands. Therefore, we invite you to submit a revised version of the manuscript that addresses the points raised during the review process.

We look forward to receiving your revised manuscript.

Kind regards,

Zhifeng Gao

Academic Editor

PLOS ONE

Journal Requirements:

Reviewers' comments:

Reviewer's Responses to Questions

**Comments to the Author**

1. Is the manuscript technically sound, and do the data support the conclusions?

Reviewer #1: Yes

Reviewer #2: Yes

2. Has the statistical analysis been performed appropriately and rigorously? 

Reviewer #1: No

Reviewer #2: Yes

3. Have the authors made all data underlying the findings in their manuscript fully available?

Reviewer #1: Yes

Reviewer #2: Yes

4. Is the manuscript presented in an intelligible fashion and written in standard English?

Reviewer #1: Yes

Reviewer #2: Yes

5. Review Comments to the Author

Reviewer #1: Overall, I found the paper to be interesting and well written.

Some of the content in the introduction could be moved to background section. For example, while I found the second paragraph to be informative and interesting, I am not how the lists of names and organizations gets a reader to the objectives of the study.

The background on Genetic Essentialism is comprehensive, the background on GM attitudes and knowledge is lacking. This should also be discussed in the conclusion to discuss how the current research aligns with previous literature. Here are just a couple of relevant papers:

1. Fernbach, P.M., Light, N., Scott, S.E., Inbar, Y. and Rozin, P., 2019. Extreme opponents of genetically modified foods know the least but think they know the most. Nature Human Behaviour, 3(3), pp.251-256.

2. McFadden, B.R. and Lusk, J.L., 2016. What consumers don't know about genetically modified food, and how that affects beliefs. The FASEB Journal, 30(9), pp.3091-3096.

I would be interested to know the median time to complete the surveys. I am guessing the first survey had at least 80 questions and the second had 110 questions? There may need to be something said about fatigue. Also, the authors could provide more information and validity of scales used; some of the scales have citations but there is no discussion about why the scales are appropriate.

In the tables, it would be good to note for readers what the endpoints of the scales indicate. Currently, there is info about the possible range of scales, but it is not clear how to interpret the means.

I expected to see some analysis completed between the two studies, particularly since the authors mention replication. There is not currently a justification for just estimating correlations; this should be made clear in the Methods section. What hypotheses can your really test? My guess is that there are more appropriate statistical methods to test hypotheses; however, the hypotheses are not clearly defined. The authors mention pre-registration, were hypotheses defined there?

It would be easier for readers if tables 1 and 2 were formatted similarly with the addition in study below (e.g., 5 could be age in both tables).

Reviewer #2: I think this paper tackles an interesting topic. It was definitely a pleasure to read. It’s beautifully written.

I have some suggestions for improvement:

Clarifying the Essentialism Construct. I thought the components or key features of genetic essentialism could be fleshed out a little more for the reader. I am sure you know them very well, but as a reader I was still not sure exactly what the key components of essentialism were. For example, there were a lot of references to “simplistic” views. What are examples of a simplistic view? Thinking one gene usually causes something complex (e.g., intelligence, a disease)? Ignoring gene X environment interactions?

Relatedly, is genetic essentialism a version of psychological essentialism? (It’s one way psychological essentialism can manifest.)

Essentialism Scales as Capturing More than Just “Essentialist Beliefs”. I have concerns that both genetic essentialism measures capture more than just the core facets of genetic essentialism. This is especially problematic given the correlational nature of the studies reported. It means that it is possible that two measures (for example, essentialism beliefs and social dominance) are correlated not because the constructs are actually correlated but because both measures are also being affected by some third construct (conservatism). More details below:

A. I could only access one scale in full (Keller 2005). I am a concerned some items may be heavily influenced (maybe even primarily determined) by other social attitudes. For example, the two items about genetic differences in race would elicit strong moral reactions in many, and seems like it is going to capture social/moral/political attitudes. Other items share a similar issue (homosexuality as a “choice” arguments come to mind for item #5; the gender differences in #3; even alcoholism and intelligence can be very complex political, social and moral topics). I list these items at the end of the review

B. It is hard for me to envision NOT getting a correlation between SDO and items like the “race” items in the scale, but not really because of genetic essentialism so much as because of social conservatism being correlated with both attitudes. It seems like it bakes in some of the correlation between SDO and GE, because of the items used.

C. Ideally, the authors could run a new study on Amazon Mechanical Turk or Prolific with a new, cleaner measure of essentialist beliefs. I actually think this would be a great contribution to the literature, if no such measure exists yet. At the very least, I think that the authors should use their current data to address these questions statistically, such as by controlling for political attitudes and demographics or choosing the most face-valid, least politicized items and using that subscale as a robustness check.

The GMO-Essentialism Correlation: The correlation with GMO attitudes is surprising. If I think one gene corresponds to one phenotypic trait, then I should be more supportive of GMO foods, right, insofar as it makes it easier to target and change the crop? I have a few thoughts:

A. Is this because science knowledge affects both essentialist attitudes and GMO attitudes? In other words, it’s possible that some people know a lot about (and trust) science. These people, because of their science knowledge and/or trust, like GMO food. These people, also because they know and trust science, disagree more with genetic essentialism items. I think it is really important to consider this as a possible confound.

B. I think controlling for science knowledge in a regression framework would shed light on some of this, and should absolutely be done. If you do another study, measuring trust in science would be helpful too.

C. I noticed the GMO finding was not preregistered. That’s completely fine, but I think it would be good to be more upfront about it if it. Just say it’s exploratory and you didn’t expect it a priori (if you didn’t)

Variability in Correlations Across Studies. You have a few measures in both studies (identical, I believe, except maybe you changed the response scales to 1-7 in study 2). The magnitude of these correlations, for a close-to-direct replication, changes quite a bit from study 1 to study 2. For example: In Study 1, the GET-eugenics acceptance correlation is .39. In Study 2 the correlation is .79. While both significant, these are pretty different effect sizes -- .79 is about what you’d expect if you were measuring the same construct with two different measures! In short, for the associations you measure in studies 1 and 2, the correlations in Study 2 look bigger (and I bet significantly so). Do you have any notion why this might be?

Other Comments:

1. Can you please post your materials as well as your data? I couldn’t find the 24-item essentialism scale.

2. There’s a small typo on p. 10 (sclae should be scale)

3. I sometimes found it tough to keep track of the abbreviations (BGD, GET) and so I might stick to spelling it out to make it easier on the reader

4. I might mention in the Study 1 methods that BGD and GET are two measures of genetic determinism, and you include both as a robustness check. I thought you were going to control for one of them until I got to the results.

Good luck! Very interesting work!

Keller Scale Items:

I believe that many differences between humans of different skin color can be attributed to differences in genetic predispositions.

I think the genetic differences between Asians and Europeans are an important cause for the differences in abilities between individuals from these groups.

I think that differences between men and women in behavior and personality are largely determined by genetic predisposition.

In my view, the development of homosexuality in a person can be attributed to genetic causes.

6. PLOS authors have the option to publish the peer review history of their article (what does this mean?). If published, this will include your full peer review and any attached files.

Reviewer #1: No

Reviewer #2: No

---

## [Author Response · Author response to Decision Letter 0]

15 May 2021

Dear Dr. Gao,

We are grateful for the opportunity to revise our manuscript and to address the critiques of the reviewers. Below, we describe our responses to each of the reviewers’ points in the order that they appear. Our responses are entered right beneath each of the reviewers’ specific comments.

Reviewer #1: Overall, I found the paper to be interesting and well written.

Some of the content in the introduction could be moved to background section. For example, while I found the second paragraph to be informative and interesting, I am not how the lists of names and organizations gets a reader to the objectives of the study.

We thank the Reviewer for this point. However, because we are dealing with a controversial topic that has largely fallen out of public view for the past 75 years, we felt it was important to provide some background information from the outset to provide a context with which to understand the goals of our paper. Our discussions with others about eugenics have revealed that most people are not aware of just how mainstream those ideas were prior to the war, and we’re trying to highlight that they were common because of the simplistic and deterministic attitudes towards genetics that were common at the time. In attempting this revision, we have tried moving this discussion to the end of the paper, however, it didn’t seem to fit well there at all. We respectfully request that we be allowed to keep this background paragraph in the introduction to provide some historical context of the link between deterministic attitudes towards genetics and support for eugenics. Indeed, this is the only recommendation from the reviewers that we wish to challenge.

The background on Genetic Essentialism is comprehensive, the background on GM attitudes and knowledge is lacking. This should also be discussed in the conclusion to discuss how the current research aligns with previous literature. Here are just a couple of relevant papers:

1. Fernbach, P.M., Light, N., Scott, S.E., Inbar, Y. and Rozin, P., 2019. Extreme opponents of genetically modified foods know the least but think they know the most. Nature Human Behaviour, 3(3), pp.251-256.

2. McFadden, B.R. and Lusk, J.L., 2016. What consumers don't know about genetically modified food, and how that affects beliefs. The FASEB Journal, 30(9), pp.3091-3096.

We thank the reviewer for suggesting these two articles. We hadn’t encountered any of them before and, in reading them now, we can see how relevant they are. We have incorporated them in the manuscript in both the introduction on pp. 5-6 where we now state “Moreover, research finds that those with the most extreme opposition to GMO foods tend to have lower knowledge about science and genetics, despite being confident in their perceived understanding of GMO foods (Fernbach, Light, Scott, Inbar, & Rozin, 2019). However, confidence in opposition to GMO products diminishes when people are pressed for their actual knowledge of what is involved in genetic modification (McFadden & Lusk, 2016),” and in the discussion on p.15 where we now state “That is, those participants who knew more about genetics tended to have more positive attitudes towards GMOs, a finding which converges with some other research (e.g., Fernbach et al., 2019).”

I would be interested to know the median time to complete the surveys. I am guessing the first survey had at least 80 questions and the second had 110 questions? There may need to be something said about fatigue. 

We calculated and the median time for the second (longer) study was 21.9 minutes, which seems in line with many other surveys. Indeed, this is one of the shorter studies that I have conducted online recently. If fatigue was an issue we’d expect to see lower reliabilities in the measure, but the alphas of the scales used in the longer study range from .87 to .97, which are all quite high.

Also, the authors could provide more information and validity of scales used; some of the scales have citations but there is no discussion about why the scales are appropriate.

We have now added on pp. 10-11 and on p. 14 some sentences that point to the validity of the scales that were used. Specifically, we now state for the measures of 1) Belief in Genetic Determinism that “This scale is a widely used measure that assesses the degree to which people believe various human characteristics are the direct product of inferred underlying genetic differences (e.g., Gould & Heine, 2012)”; 2) Genetic Essentialist Tendencies that “This scale builds from the genetic essentialist framework from Dar-Nimrod and Heine (2011) and assesses the degree to which people view genetic causes in essentialist terms. Previous research suggests it correlates strongly with the Belief in Genetic Determinism measure (e.g., Schmalor, Cheung, & Heine, 2021)”; Social Dominance Orientation that “This widely used scale assesses the degree to which people view some social groups to be superior to others”; Attitudes towards Genetically-Modified food that “This scale has been used in previous research investigating the bases of people’s support and opposition to GMO foods (e.g., Royzman, Cusimano, & Leeman, 2017)”; and Knowledge about Genetics that “These measures of genetic knowledge have been used in previous research (e.g., Molster, Charles, Samanek, & O’Leary, 2009; Schmalor et al., 2021).”

In the tables, it would be good to note for readers what the endpoints of the scales indicate. Currently, there is info about the possible range of scales, but it is not clear how to interpret the means.

We have now added the scale endpoints to both tables.

I expected to see some analysis completed between the two studies, particularly since the authors mention replication. There is not currently a justification for just estimating correlations; this should be made clear in the Methods section. What hypotheses can your really test? My guess is that there are more appropriate statistical methods to test hypotheses; however, the hypotheses are not clearly defined. The authors mention pre-registration, were hypotheses defined there?

We thank the reviewer for this request. We have now highlighted explicitly in the text on p. 10 that our plan was to analyze the correlations among the relevant dependent variables. While the first study was exploratory the second study was confirmatory, and we spelled out the following hypotheses in the pre-registration:

Hypothesis 1a: People who have less knowledge about genes support eugenics more than do people who have more knowledge. 

Hypothesis 1b: People who score higher on scales of genetic essentialism support eugenics more than people who score lower. 

Hypothesis 2: People who have less knowledge about genes score higher on scales of genetic essentialism than do people who have more knowledge. 

Hypothesis 3: People who score higher on scales of genetic essentialism score higher on social dominance orientation. 

It would be easier for readers if tables 1 and 2 were formatted similarly with the addition in study below (e.g., 5 could be age in both tables).

We have now changed the order of the variables so that they are identical between Tables 1 and 2.

Reviewer #2: I think this paper tackles an interesting topic. It was definitely a pleasure to read. It’s beautifully written.

I have some suggestions for improvement:

Clarifying the Essentialism Construct. I thought the components or key features of genetic essentialism could be fleshed out a little more for the reader. I am sure you know them very well, but as a reader I was still not sure exactly what the key components of essentialism were. For example, there were a lot of references to “simplistic” views. What are examples of a simplistic view? Thinking one gene usually causes something complex (e.g., intelligence, a disease)? Ignoring gene X environment interactions?

Relatedly, is genetic essentialism a version of psychological essentialism? (It’s one way psychological essentialism can manifest.)

We thank the reviewer for calling attention to our sparse description of essentialism. We have now expanded our discussion of genetic essentialism on pp. 6-7, where we now provide the following paragraph: “Dar-Nimrod and Heine (2011) argue that there are four key aspects of genetic essentialist thinking. First, genetic causes tend to be perceived as immutable; people may view that someone who possesses a gene linked to a particular condition will inevitably develop that condition. Second, when people make genetic attributions for a phenomenon, they often discount other kinds of potential causes so that genetic factors may be seen as the ultimate cause. A third feature of genetic essentialist thinking is that people tend to view groups that share genes as more homogenous, and more discrete from groups that possess alternative alleles; that is, genes are perceived to serve as the basis of natural categories. Last, genes tend to be perceived as natural causes, and, as such, any interventions to modify them may strike people as deeply unnatural, and bothersome.” 

Essentialism Scales as Capturing More than Just “Essentialist Beliefs”. I have concerns that both genetic essentialism measures capture more than just the core facets of genetic essentialism. This is especially problematic given the correlational nature of the studies reported. It means that it is possible that two measures (for example, essentialism beliefs and social dominance) are correlated not because the constructs are actually correlated but because both measures are also being affected by some third construct (conservatism). More details below:

A. I could only access one scale in full (Keller 2005). I am a concerned some items may be heavily influenced (maybe even primarily determined) by other social attitudes. For example, the two items about genetic differences in race would elicit strong moral reactions in many, and seems like it is going to capture social/moral/political attitudes. Other items share a similar issue (homosexuality as a “choice” arguments come to mind for item #5; the gender differences in #3; even alcoholism and intelligence can be very complex political, social and moral topics). I list these items at the end of the review

B. It is hard for me to envision NOT getting a correlation between SDO and items like the “race” items in the scale, but not really because of genetic essentialism so much as because of social conservatism being correlated with both attitudes. It seems like it bakes in some of the correlation between SDO and GE, because of the items used.

C. Ideally, the authors could run a new study on Amazon Mechanical Turk or Prolific with a new, cleaner measure of essentialist beliefs. I actually think this would be a great contribution to the literature, if no such measure exists yet. At the very least, I think that the authors should use their current data to address these questions statistically, such as by controlling for political attitudes and demographics or choosing the most face-valid, least politicized items and using that subscale as a robustness check.

We thank the Reviewer for raising this point. We feel the current data that we have allows us to test their hypothesis that the correlations with SDO are driven by the potentially problematic items that they list from the Belief in Genetic Determinism measure. In our new analysis we have removed the items that the reviewer identified as potentially problematic and we have recalculated all of the correlations with this new scale for Study 1. The alpha for the shortened scale (.85) was essentially the same as for the complete scale (.86). We paste below how the two tables of correlations compare for the shortened vs. the original scale. As you can see, while the correlations were nominally smaller with the shortened scale, all of the significant correlations remained significant at p < .001, indicating that the significant correlations were not due to the specific items that the reviewer identified. As such, we have opted not to change the scale that we used in the manuscript to keep it in line with other published research that relies on this scale.

 

Shorter BGD

Vs original:

The GMO-Essentialism Correlation: The correlation with GMO attitudes is surprising. If I think one gene corresponds to one phenotypic trait, then I should be more supportive of GMO foods, right, insofar as it makes it easier to target and change the crop? I have a few thoughts:

A. Is this because science knowledge affects both essentialist attitudes and GMO attitudes? In other words, it’s possible that some people know a lot about (and trust) science. These people, because of their science knowledge and/or trust, like GMO food. These people, also because they know and trust science, disagree more with genetic essentialism items. I think it is really important to consider this as a possible confound.

We hadn’t hypothesized about the relation between GMO and essentialism, however, to us it makes sense that people with more essentialistic attitudes towards genetics should be especially bothered by GMOs. As we now note in the intro on p. 7 that a key genetic essentialist intuition is that genetic causes are natural, so that any planned alteration of genes may strike people as deeply unnatural, and bothersome.

B. I think controlling for science knowledge in a regression framework would shed light on some of this, and should absolutely be done. If you do another study, measuring trust in science would be helpful too.

We agree with the reviewer that it would be interesting to explore trust in science, however, we feel this is a somewhat tangentially-related question that would be best left for future research.

C. I noticed the GMO finding was not preregistered. That’s completely fine, but I think it would be good to be more upfront about it if it. Just say it’s exploratory and you didn’t expect it a priori (if you didn’t)

We now specify explicitly on p. 15 that the finding with GMO was not preregistered as we now say “In addition, although not preregistered, we found that knowledge about genetics was positively correlated with attitudes towards GMOs, r=.24, p<.001”

Variability in Correlations Across Studies. You have a few measures in both studies (identical, I believe, except maybe you changed the response scales to 1-7 in study 2). The magnitude of these correlations, for a close-to-direct replication, changes quite a bit from study 1 to study 2. For example: In Study 1, the GET-eugenics acceptance correlation is .39. In Study 2 the correlation is .79. While both significant, these are pretty different effect sizes -- .79 is about what you’d expect if you were measuring the same construct with two different measures! In short, for the associations you measure in studies 1 and 2, the correlations in Study 2 look bigger (and I bet significantly so). Do you have any notion why this might be?

We agree with the Reviewer that these two correlations are surprisingly different. We double-checked and these values are indeed correct. We don’t have any well-formed thoughts on why they are so different, and we were careful not to draw any conclusions in the paper regarding the magnitude of the respective correlations.

Other Comments:

1. Can you please post your materials as well as your data? I couldn’t find the 24-item essentialism scale.

We have now added the 24-item essentialism scale to the Appendix. All of the scales are now available in the appendix or are referenced (i.e., Keller, 2005; Pratto et al., 1994; Scott et al., 2016).

2. There’s a small typo on p. 10 (sclae should be scale)

We have corrected this now.

3. I sometimes found it tough to keep track of the abbreviations (BGD, GET) and so I might stick to spelling it out to make it easier on the reader

We agree that these acronyms can be challenging to keep track of have replaced BGD and GET with Beliefs in Genetic Determinism and Genetic Essentialist Tendencies throughout the manuscript now.

4. I might mention in the Study 1 methods that BGD and GET are two measures of genetic determinism, and you include both as a robustness check. I thought you were going to control for one of them until I got to the results.

We now say on p. 12 that “We included two distinct measures of genetic essentialist biases as a test of robustness.”

Good luck! Very interesting work!

Keller Scale Items:

I believe that many differences between humans of different skin color can be attributed to differences in genetic predispositions.

I think the genetic differences between Asians and Europeans are an important cause for the differences in abilities between individuals from these groups.

I think that differences between men and women in behavior and personality are largely determined by genetic predisposition.

In my view, the development of homosexuality in a person can be attributed to genetic causes.

We thank the reviewers for their helpful comments and their kind words about our research.

On Behalf of the Authors,

---

## [Decision Letter · Decision Letter 1]

1 Jul 2021

PONE-D-21-03521R1

The Role of Genetic Essentialism and Genetics Knowledge in Support for Eugenics and Genetically Modified Foods

PLOS ONE

Dear Dr. Heine,

Thank you for submitting your manuscript to PLOS ONE. After careful consideration, we feel that it has merit but does not fully meet PLOS ONE’s publication criteria as it currently stands. Therefore, we invite you to submit a revised version of the manuscript that addresses the points raised during the review process.

Please carefully address reviewer 2's comments related to the instruments and data description. These comments are very specific but essential. Thanks.  

We look forward to receiving your revised manuscript.

Kind regards,

Zhifeng Gao

Academic Editor

PLOS ONE

Journal Requirements:

Reviewers' comments:

Reviewer's Responses to Questions

**Comments to the Author**

1. If the authors have adequately addressed your comments raised in a previous round of review and you feel that this manuscript is now acceptable for publication, you may indicate that here to bypass the “Comments to the Author” section, enter your conflict of interest statement in the “Confidential to Editor” section, and submit your "Accept" recommendation.

Reviewer #1: All comments have been addressed

Reviewer #2: (No Response)

2. Is the manuscript technically sound, and do the data support the conclusions?

Reviewer #1: Yes

Reviewer #2: Partly

3. Has the statistical analysis been performed appropriately and rigorously? 

Reviewer #1: Yes

Reviewer #2: Yes

4. Have the authors made all data underlying the findings in their manuscript fully available?

Reviewer #1: Yes

Reviewer #2: Yes

5. Is the manuscript presented in an intelligible fashion and written in standard English?

Reviewer #1: Yes

Reviewer #2: Yes

6. Review Comments to the Author

Reviewer #1: Thank you for addressing my comments. Nice work!

There is a character limit for this box, so ignore this.

Reviewer #2: Here are some comments on the current version of the manuscript.

I have one new comment on the GMO Attitudes Measure:

You write: “Participants completed the 9-item measure from Scott et al., (2016). Five of these items asked about support of GMO products (e.g., “Your government forbidding imports of genetically modified (GM) foods from other countries” and participants indicated their support on a 9-point scale from 1 (Certainly oppose) to 9 (Certainly support). Four of these items asked about perceived risk of GMO products (e.g., Genetically modified foods having unknown side-effects, increasing risks of cancer or other diseases for people who consume them” which participants answered on a 9-point scale from 1 (No risk at all) to 9 (Extreme risk). The latter subscale was reverse coded so that higher scores meant more favorable attitudes. This scale has been used in previous research investigating the bases of people’s support and opposition to GMO foods (e.g., Royzman, Cusimano, & Leeman, 2017). Cronbach’s α = 0.93. "

In the first example item, indicating 9 = certainly support means you support forbidding imports of GMO foods, meaning you oppose GMO foods. You say you only reverse score the risk items and then average all the items together. Is that true? If so, that’s a major issue, because you are averaging together 5 (not reverse-scored) items where high scores indicate opposition and 4 (reverse-scored) items where high scores indicate support. I’m pretty sure it’s just a mistake in the writeup, looking at your code and the data. If it is a mistake in the writeup, please confirm that all items were reverse coded and rewrite this methods paragraph. If this was done on purpose, please explain more thoroughly why.

Also, while looking at the data file/code to try to understand this, I noticed you only have 3 gmo risk items in your data file, not four as stated in the methods section.

Relatedly, the original endpoints in the Table 2 can be confusing. It makes it look like GMO attitudes is a measure where higher scores = more negative attitudes, but you’ve reverse coded it.

Why not rename the GMO attitudes measure so it’s clear what direction it goes in throughout the manuscript (for example: “Positive Attitudes to GMOs” or perhaps “GMO Acceptance”, as you do with your “eugenics acceptance” measure)?

A number of items from my last review that have not been fully addressed.

1. (Essentialism Items as Capturing More than Just Essentialist Beliefs from last round): I cannot see the tables in your letter, which I assume is an issue with copying and pasting into the journal’s web interface. All I see is “Shorter BGD Vs original: ”. Can you please include those correlations in the next response letter?

2. (GMO-Essentialism Correlation point last round): I think the correlational nature and the issue of third variables of trust in science/knowledge should be more fully acknowledged in the limitations section than just a sentence or two. When I see the following findings--people who support eugenics are more negative to GMOs and that people who have more simplistic/deterministic views of genetics causes (essentialism) are more negative to GMOs--the first thing I wonder is whether this is all driven by some type of general trust in science or broad science education. This needs to be addressed in the manuscript.

3. (#4 in Other Comments): Can you move the clarification you added in the study 1 results section up to the study 1 methods section about BGD and GET being two measures capturing the same thing? I still think it is confusing when you read the BGD and GET paragraphs.

4. Variability in Correlations Across Studies. I think it matters that your effect sizes change a LOT in a near-direct replication. This should be acknowledged in the manuscript. One possibility is TurkPrime recruits more attentive MTurk workers, which reduces noise, but this probably doesn’t account for the entire difference.

Other: (page numbers from track changes version):

p. 2 “highlighting the value of education in genetics” doesn’t this phrase really go beyond the scope of your data, and imply a causal relationship?

p. 11 “24 item” should be 24-item; sclae should be scale in genetic essentialist tendencies too

p. 14 7 point should be 7-point

7. PLOS authors have the option to publish the peer review history of their article (what does this mean?). If published, this will include your full peer review and any attached files.

Reviewer #1: No

Reviewer #2: No

---

## [Author Response · Author response to Decision Letter 1]

12 Jul 2021

Dear Dr. Gao,

We are grateful for the opportunity to revise our manuscript and to address the critiques of the reviewers. Below, we describe our responses to each of the reviewers’ points in the order that they appear. Our responses are entered right beneath each of the reviewers’ specific comments.

Reviewer #1: Thank you for addressing my comments. Nice work!

There is a character limit for this box, so ignore this.

We thank the reviewer for their positive assessment.

Reviewer #2: Here are some comments on the current version of the manuscript.

I have one new comment on the GMO Attitudes Measure:

You write: “Participants completed the 9-item measure from Scott et al., (2016). Five of these items asked about support of GMO products (e.g., “Your government forbidding imports of genetically modified (GM) foods from other countries” and participants indicated their support on a 9-point scale from 1 (Certainly oppose) to 9 (Certainly support). Four of these items asked about perceived risk of GMO products (e.g., Genetically modified foods having unknown side-effects, increasing risks of cancer or other diseases for people who consume them” which participants answered on a 9-point scale from 1 (No risk at all) to 9 (Extreme risk). The latter subscale was reverse coded so that higher scores meant more favorable attitudes. This scale has been used in previous research investigating the bases of people’s support and opposition to GMO foods (e.g., Royzman, Cusimano, & Leeman, 2017). Cronbach’s α = 0.93. "

In the first example item, indicating 9 = certainly support means you support forbidding imports of GMO foods, meaning you oppose GMO foods. You say you only reverse score the risk items and then average all the items together. Is that true? If so, that’s a major issue, because you are averaging together 5 (not reverse-scored) items where high scores indicate opposition and 4 (reverse-scored) items where high scores indicate support. I’m pretty sure it’s just a mistake in the writeup, looking at your code and the data. If it is a mistake in the writeup, please confirm that all items were reverse coded and rewrite this methods paragraph. If this was done on purpose, please explain more thoroughly why.

We thank the Reviewer for catching this mistake. We did in fact reverse-code all items, not just the risk items, so that higher scores indicate more support for GMO foods. We have now corrected this on page 14 where we now say “Both subscales were reverse coded so that higher scores meant more favorable attitudes towards GMO foods.”

Also, while looking at the data file/code to try to understand this, I noticed you only have 3 gmo risk items in your data file, not four as stated in the methods section.

We thank the Reviewer for catching yet another mistake. We apologize for our carelessness. Yes, we only included 3 GMO risk items in the study. We excluded one of the four original items which did not refer to an actual genetic risk, but a risk of big corporations having too much power, as this seemed to us to bring up a rather unrelated kind of concern of the business regarding GMO products. We have added a footnote in the manuscript now which says “The Scott et al. paper included an additional item about GMO risks that read ‘Genetically modified crops giving big corporations too much power over small farmers.’ We omitted this item from our study as it did not seem to tap into fears of genetic modifications per se, but concerns about implications of business practices regarding GMO products.”

Relatedly, the original endpoints in the Table 2 can be confusing. It makes it look like GMO attitudes is a measure where higher scores = more negative attitudes, but you’ve reverse coded it.

Why not rename the GMO attitudes measure so it’s clear what direction it goes in throughout the manuscript (for example: “Positive Attitudes to GMOs” or perhaps “GMO Acceptance”, as you do with your “eugenics acceptance” measure)?

We thank the Reviewer for this excellent point and we have changed the paper accordingly. We now refer to the GMO measure as “GMO Acceptance,” both in the Table, and in the Results section. Also, in Table 2 we have removed the labels for the original endpoints as those only refer to the risk items, and given that the entire scale is reverse-scored, we agree with the reviewer that this only increases confusion.

A number of items from my last review that have not been fully addressed.

1. (Essentialism Items as Capturing More than Just Essentialist Beliefs from last round): I cannot see the tables in your letter, which I assume is an issue with copying and pasting into the journal’s web interface. All I see is “Shorter BGD Vs original: ”. Can you please include those correlations in the next response letter?

There must have been an issue with how the correlations showed up in the letter on the online portal as the two tables of the correlations were indeed there on the original letter that we submitted. We include the two tables again here. In case it remains invisible to the Reviewer again we note that the shorter BGD measure correlated .233 with eugenic acceptance, .589 with the GET, .122 with SDO, and .118 with age – all of these are significant at p < .001, as they were with the original scale. Gender did not correlate with either BGD scale (-.007 with the shorter measure, and -.044 with the longer one). The alpha of the shorter scale is .85 in comparison to .86 for the entire scale.

Shorter BGD

Vs original:

2. (GMO-Essentialism Correlation point last round): I think the correlational nature and the issue of third variables of trust in science/knowledge should be more fully acknowledged in the limitations section than just a sentence or two. When I see the following findings--people who support eugenics are more negative to GMOs and that people who have more simplistic/deterministic views of genetics causes (essentialism) are more negative to GMOs--the first thing I wonder is whether this is all driven by some type of general trust in science or broad science education. This needs to be addressed in the manuscript.

We agree with the Reviewer about the limits of the correlational data, and we have added the following sentence to the Limitations Section on page 17. “In particular, it is possible that a third variable, such as general trust in science, cognitive style, or broad science education, may underlie the correlations of some of these variables, including attitudes towards GMOs, eugenics acceptance, and genetics knowledge.”

3. (#4 in Other Comments): Can you move the clarification you added in the study 1 results section up to the study 1 methods section about BGD and GET being two measures capturing the same thing? I still think it is confusing when you read the BGD and GET paragraphs.

We have now moved the sentence to the method section on page 11 which specifies “We included two distinct measures of genetic essentialist biases as a test of robustness.”

4. Variability in Correlations Across Studies. I think it matters that your effect sizes change a LOT in a near-direct replication. This should be acknowledged in the manuscript. One possibility is TurkPrime recruits more attentive MTurk workers, which reduces noise, but this probably doesn’t account for the entire difference.

We have now added a sentence on page 15 to acknowledge the difference in the magnitude in the correlations between the two studies, which reads: “Unexpectedly, these correlations were considerably larger than they were in Study 1, and we are not sure what aspect of the sample or survey design might be responsible for these differences in magnitude.”

Other: (page numbers from track changes version):

p. 2 “highlighting the value of education in genetics” doesn’t this phrase really go beyond the scope of your data, and imply a causal relationship?

We agree that this sentence implies a causal relationship, which to us seems like the most reasonable take on the underlying nature of the correlation, although we acknowledge that third variables very well may be responsible. We have slightly toned this down by adding the word “potentially” before highlighting.

p. 11 “24 item” should be 24-item; sclae should be scale in genetic essentialist tendencies too

p. 14 7 point should be 7-point

We have made these minor corrections now.

We thank the reviewers for their helpful comments and their kind words about our research.

On Behalf of the Authors,

Steve Heine

---

## [Decision Letter · Decision Letter 2]

15 Sep 2021

The Role of Genetic Essentialism and Genetics Knowledge in Support for Eugenics and Genetically Modified Foods

PONE-D-21-03521R2

Dear Dr. Heine,

We’re pleased to inform you that your manuscript has been judged scientifically suitable for publication and will be formally accepted for publication once it meets all outstanding technical requirements.

Please consider the reviewer's last comment when submitting your final version. For your convenience, the comment is provided here

"I just have a very small clarity suggestion. You moved this sentence to the end of the Genetic Essentialist Tendencies paragraph: “We included two distinct measures of genetic essentialist biases as a test of robustness.” I think this would be clearer: “We included two distinct measures of genetic essentialist biases (the Belief in Genetic Determinism Scale and the Genetic Essentialist Tendencies Scale) as a test of robustness.” Otherwise, a reader might wonder if they missed that there are two measures being described in the Genetic Essentialist Tendencies paragraph."

Kind regards,

Zhifeng Gao

Academic Editor

PLOS ONE

Additional Editor Comments (optional):

Reviewers' comments:

Reviewer's Responses to Questions

**Comments to the Author**

1. If the authors have adequately addressed your comments raised in a previous round of review and you feel that this manuscript is now acceptable for publication, you may indicate that here to bypass the “Comments to the Author” section, enter your conflict of interest statement in the “Confidential to Editor” section, and submit your "Accept" recommendation.

Reviewer #2: All comments have been addressed

2. Is the manuscript technically sound, and do the data support the conclusions?

Reviewer #2: Yes

3. Has the statistical analysis been performed appropriately and rigorously? 

Reviewer #2: Yes

4. Have the authors made all data underlying the findings in their manuscript fully available?

Reviewer #2: Yes

5. Is the manuscript presented in an intelligible fashion and written in standard English?

Reviewer #2: Yes

6. Review Comments to the Author

Reviewer #2: Thanks for addressing the comments. I just have a very small clarity suggestion. You moved this sentence to the end of the Genetic Essentialist Tendencies paragraph: “We included two distinct measures of genetic essentialist biases as a test of robustness.” I think this would be clearer: “We included two distinct measures of genetic essentialist biases (the Belief in Genetic Determinism Scale and the Genetic Essentialist Tendencies Scale) as a test of robustness.” Otherwise, a reader might wonder if they missed that there are two measures being described in the Genetic Essentialist Tendencies paragraph.

Good job!

7. PLOS authors have the option to publish the peer review history of their article (what does this mean?). If published, this will include your full peer review and any attached files.

Reviewer #2: No

---

## [Editor Report · Acceptance letter]

22 Sep 2021

PONE-D-21-03521R2 

The Role of Genetic Essentialism and Genetics Knowledge in Support for Eugenics and Genetically Modified Foods 

Dear Dr. Heine:

I'm pleased to inform you that your manuscript has been deemed suitable for publication in PLOS ONE. Congratulations! Your manuscript is now with our production department. 

Kind regards, 

on behalf of

Dr. Zhifeng Gao 

Academic Editor

PLOS ONE